# Transcranial Direct Current Stimulation Enhances Motor Performance by Modulating Beta-Phase Synchronization in the Sensorimotor Network: A Preliminary Study

**DOI:** 10.3390/brainsci15030286

**Published:** 2025-03-07

**Authors:** Eri Miyauchi, Yoshiki Henmi, Masahiro Kawasaki

**Affiliations:** 1Institute of Systems and Information Engineering, University of Tsukuba, Tsukuba 305-8533, Japan; 2Graduate School of Systems and Information Engineering, University of Tsukuba, Tsukuba 305-8533, Japan

**Keywords:** EEG, oscillatory neural network, phase synchronization, tDCS, tACS, beta, sensorimotor

## Abstract

Background/Objectives: Synchronized beta-band oscillations (14–30 Hz) are critical for sensorimotor processing and motor performance. Modulating beta activity either locally in targeted brain regions or globally across sensorimotor networks may enhance motor function. This study aimed to explore whether transcranial direct current stimulation (tDCS) and alternating current stimulation (tACS) could enhance sensorimotor responses by modulating beta-band synchronization. Methods: Eight participants performed a stimulus–response task requiring a quick keypress to a visual cue. Response times (RTs) and electroencephalography (EEG) data were recorded during pre-, in-, and post-stimulation sessions for five conditions: motor-anodal tDCS, visual-anodal tDCS, alpha (10 Hz) tACS, beta (20 Hz) tACS, and sham, with a one-week interval between conditions. Results: Significant RT reductions were observed only after motor-anodal tDCS. EEG analysis revealed a positive correlation between these RT reductions and increased beta-phase synchronization between visual and motor areas. In contrast, tACS conditions did not yield significant RT improvements or beta-phase synchronization changes. Conclusions: These findings indicate that motor-anodal tDCS has the potential to enhance sensorimotor performance by facilitating beta-phase synchronization across the visual-motor network. The observed effects likely extend beyond localized neuronal modulation, emphasizing the importance of network-level connectivity in sensorimotor integration. Beta-phase synchronization appears to play a critical role in integrating visual and motor information, contributing to task-related performance improvements. Further research is warranted to build upon these findings and fully elucidate the underlying mechanisms.

## 1. Introduction

The modulation of brain activity to maintain or improve motor function has been a major topic in neuroscience. Neuroimaging studies have advanced the practicality of this research by elucidating the relevant neural areas, networks, and underlying mechanisms. For instance, functional magnetic resonance imaging (fMRI) and positron emission tomography studies have revealed that motor function is associated with brain regions such as the primary motor cortex, supplementary motor area, and cerebellar cortex [1,2]. These brain regions are involved in the sensorimotor systems, which process external stimuli, interpret internal sensations, evaluate sensory information, and generate motor responses. Moreover, electroencephalography (EEG) studies have linked beta-band (14–30 Hz) oscillations with motor function, as the beta-band amplitude in motor areas is modulated in relation to voluntary movement [3] or increased around the time of stimulus presentation, followed by a decrease during movement execution [4]. Furthermore, synchronized beta-band oscillations in large-scale brain networks have been associated with sensorimotor processing. For instance, beta-band synchronization between the visual and motor areas increases during visuomotor processing [5], visual attention [6], and visual stimulus expectancy [7]. In addition, phase synchronization between sensory and motor areas has been observed during sensorimotor tasks [8,9]. In particular, our previous study demonstrated that beta-phase synchronization increases in the frontal–temporal–cerebellar network during auditory–motor learning tasks [10], suggesting the potential effectiveness of modulating beta-band oscillatory activity in a sensorimotor network to enhance relevant motor functions.

In recent years, non-invasive brain stimulation techniques, such as transcranial magnetic stimulation (TMS) and transcranial electric stimulation (tES), have been extensively utilized in neuroenhancement studies. Particular interest has been directed toward transcranial direct current stimulation (tDCS), a major form of tES, which uses a weak electric current delivered via electrodes placed on the scalp to modulate local neuronal activity [11,12,13]. tDCS can induce a long-term neural enhancement and suppression in anodal and cathodal brain areas, respectively [14]. Weak anodal tDCS to motor areas enhances motor speed [15], accuracy [16], memory [17], and learning [18], whereas cathodal tDCS to motor areas exerts no effect on motor performance [15,19], suggesting that anodal stimulation has greater benefits for motor function enhancement.

When currents are applied alternately (transcranial alternating current stimulation [tACS]), it is assumed that electric currents at specific frequencies are induced on the applied electrodes, thereby modulating cognition and behaviors [20,21]. For instance, beta (20 Hz) tACS has been reported to induce 20 Hz oscillations and facilitate performance in both motor [22] and cognitive tasks [23].

Although tES is considered to be a promising neuroenhancement technology, its efficacy has shown mixed results [12,24], underscoring the need for a deeper understanding of its underlying mechanisms. Notably, few studies have investigated the efficacy of tES in enhancing motor performance by analyzing the synchronized beta-band oscillatory activity within relevant cortical networks. Moreover, many of these studies have focused exclusively on either tDCS or tACS, making it challenging to directly compare the modulatory effects of these methods. The local entrainment of oscillatory activity may be particularly useful for examining the activity of targeted areas, especially in tDCS, whereas the global entrainment of oscillatory activity across distant relevant areas could provide insights into network-level effects, particularly in the case of tACS.

Therefore, this study primarily aimed to examine the effects of targeted motor-anodal tDCS and beta-tACS on motor performance and neural activity, as these two conditions were hypothesized to induce localized and network-level modulation, respectively. To assess the specificity of the observed effects, additional conditions—anodal tDCS over the visual cortex and alpha-tACS—were included as comparisons. These conditions allowed us to determine whether the effects of motor-anodal tDCS and beta-tACS were frequency- and region-specific. The sham condition was used as a baseline to account for non-stimulation effects. To achieve this, a within-subject design was employed, and EEG data were collected while participants performed a stimulus–response task during pre-, in-, and post-stimulation sessions. Each stimulation condition was separated by a minimum interval of one week to mitigate potential carryover effects.

Based on the hypothesized mechanisms of action and the findings from previous tES studies, it was anticipated that tDCS would enhance motor performance by increasing local synchronization in the targeted area. Specifically, anodal tDCS over the motor cortex was expected to increase the beta amplitude in the motor area. In contrast, tACS was hypothesized to facilitate motor performance by enhancing task-relevant sensorimotor network connectivity. Specifically, beta-frequency tACS was expected to increase the beta-phase synchronization between the motor and visual areas.

## 2. Materials and Methods

### 2.1. Experimental Design

#### 2.1.1. Participants

Eight right-handed participants (four women and four men; age range, 21–24 years; mean age, 23.7 years) with normal or corrected-to-normal vision and normal motor function completed this study.

#### 2.1.2. Task Procedure

The participants performed a stimulus-detection task on a computer display. During the task, one red square and three gray squares were simultaneously presented in a square formation surrounding a central fixation cross (top right, top left, bottom right, and bottom left; Figure 1A). The location of the red square was pseudorandomly chosen for each trial, and participants were instructed to detect the red square and respond as quickly as possible using a different finger and keypress for corresponding location (i.e., top right: “K” and right middle finger; top left: “G” and left middle finger; bottom right: “J” and right index finger; bottom left: “H” and left index finger). Motor performance was evaluated based on the response time (RT). If participants did not respond within 1.5 s, the next trial appeared. Inter-trial intervals were randomly jittered between 0.5 and 3.0 s. Participants were required to fixate their eyes on a central fixation cross throughout the task.

#### 2.1.3. Experimental Procedure

All participants completed a series of experiments involving five stimulation conditions (Figure 1B): (1) anodal tDCS over the motor cortex (motor-anodal tDCS), (2) anodal tDCS over the visual cortex (visual-anodal tDCS), (3) tACS at an alpha frequency (10 Hz; alpha-tACS), (4) tACS at a beta frequency (20 Hz; beta-tACS), and (5) a sham condition (sham). The primary focus was on the motor-anodal tDCS and beta-tACS conditions, as these were hypothesized to modulate motor performance and relevant neural activity. The visual-anodal tDCS and alpha-tACS conditions were included as comparison conditions to assess whether the effects of motor-anodal tDCS and beta-tACS were specific to stimulation location and frequency. The sham condition served as a non-stimulated baseline.

Inter-conditional intervals were created for at least one week to prevent the practice effect, and the order of the conditions was counterbalanced across participants. Each experiment consisted of three sessions: pre-, in-, and post-stimulation. Each session consisted of two blocks of 96 trials (four response types × 24 trials). The pre-stimulation session served as a baseline for comparison.

#### 2.1.4. Transcranial Electrical Stimulation

For this study, tDCS/tACS were applied using two Ag/AgCl electrodes (DC-STIMULATOR; NeuroCare Group, Munich, Germany). The intensity of the electric current was 100 μA (0.88 mA/cm^2^), and the duration of stimulation was 20 min (1.06 C/cm^2^). Under both tDCS/tACS conditions, the electrodes were placed over the motor (C3) and visual (Oz) areas in accordance with the international 10–20 EEG system. For motor-anodal tDCS stimulation, the anodal electrode was placed over the motor area, whereas, for visual-anodal tDCS stimulation, the anodal electrode was placed over the visual area. To evaluate both local neural changes and the functional connectivity between the visual cortex (Oz) and the left motor cortex (C3), our tDCS montage was deliberately configured with the anodal electrode placed on one site and the cathodal electrode on the other. This configuration was chosen to model a current flow between these areas and to capture both amplitude changes and phase synchronization. In addition, tACS was applied to both electrodes because we considered that alternating current may more effectively promote phase synchronization. For the tACS conditions, stimulation was applied as an alternating sinusoidal wave at a frequency of 10 Hz (for alpha) or 20 Hz (for beta). In the sham condition, the electrodes were positioned as in active conditions, with a 30 s ramp-up and 30 s ramp-down applied at the start, and no further current delivered.

#### 2.1.5. Electroencephalography Recording

Electroencephalography was recorded continuously during task performance using scalp electrodes (Ag/AgCl) placed over the motor (C3) and visual (Oz) areas, following the extended version of the international 10–20 system. EEG data were obtained at a sampling rate of 1000 Hz using the SynAmp2 software scan 4.5 (Neuroscan, El Paso, TX, USA). Wireless reference electrodes were placed on the left and right mastoids. Four electrodes were used to record eye blinks and movements. Electrodes were placed above and below the left eye to monitor eye blinks and vertical eye movements and 1 cm from the right and left eyes to monitor horizontal eye movements. EEG data were bandpass-filtered between 0.1 and 30 Hz.

### 2.2. Analyses

All analyses were performed using MATLAB (R2015b; MathWorks Inc., Natick, MA, USA).

#### 2.2.1. Behavioral Analysis

Motor performance was evaluated based on RTs following stimulus onset. Trials in which the RTs were above one standard deviation from the average individual RTs were removed as outliers. The results were nearly identical when the other standard deviations (e.g., 2 or 2.5) were used. The averaged skewness and kurtosis of the RTs were 0.67 ± 0.05 and 1.92 ± 0.05, respectively, indicating that the data were not normally distributed. Therefore, non-parametric analyses were conducted.

To assess stimulation effects at the individual level, z-values were calculated for each session and condition using the Wilcoxon signed-rank test. Negative z-values indicated that RTs were shorter in the in/post-stimulation session compared with the pre-stimulation session. Subsequently, stimulation effects across participants were analyzed. The total z-values were calculated using the Wilcoxon signed-rank test to determine whether the mean distribution of individual z-values was zero. The significantly negative total z-value indicated that shorter RTs were observed among participants in the in/post-stimulation session than in the pre-stimulation session.

#### 2.2.2. Amplitude Analysis

Oscillatory brain activity was assessed using recorded EEG data. Trials were excluded from analysis if the amplitude of any electrode exceeded ±100 µV during the 100 ms prior to stimulus onset or the 400 ms following stimulus onset. Owing to the inability to completely eliminate electrical artifacts in the in-stimulation session data, the analysis focused on data from the pre- and post-stimulation sessions. The modulatory effects of stimulation on local oscillatory activity were evaluated based on the time–frequency amplitudes of the motor (C3) and visual (Oz) electrodes.

The time–frequency amplitude was calculated with wavelet transforms using the Morlet wavelet function $w\left(t,f_{0}\right)$ [25]. Morlet wavelets $w\left(t,f_{0}\right)$ have a Gaussian shape around their central frequency $f_{0}$ in both the time domain (SD $\sigma_{t}$) and frequency domain (SD $\sigma_{f}$).$$w\left(t,f\right)={\left(\sigma_{t}\sqrt{\pi }\right)}^{-\frac{1}{2}}\exp\left(-\frac{t^{2}}{2\sigma_{t}^{2}}\right)\exp\left(i2\pi ft\right)$$$$E\left(t,f\right)={\left|w(t,f)⊗s(t)\right|}^{2}$$
where σf=12πσt.

A wavelet that was characterized by a constant ratio fσf=7 was used, with f ranging from 1 to 30 Hz (0.5 Hz steps). The time–frequency amplitude $E\left(t,f_{0}\right)$ for each time point of each trial was the square norm of the convolution of a complex wavelet $w\left(t,f_{0}\right)$ with the original EEG signals $s\left(t\right)$. The time–frequency amplitude for each epoch was calculated 1.5 s before and after the stimulus onset. The time–frequency amplitudes were segmented into two frequencies: alpha (10 Hz) and beta (20 Hz).

Subsequently, individual z-values were calculated using the Wilcoxon signed-rank test to compare the EEG amplitudes between the pre- and post-stimulation sessions for each condition. Positive z-values indicated that the EEG amplitude values were greater in the post-stimulation session than in the baseline session.

The correlations between the differences in RTs and EEG amplitudes between the pre- and post-stimulation sessions for all conditions were analyzed using Spearman’s correlation coefficient to determine the relationship between the stimulation effects on motor performance and the modulatory effects of stimulation on local target areas. Statistical differences were evaluated using the Mann–Whitney U test.

#### 2.2.3. Phase Synchronization Analysis

The modulatory effects of stimulation on global oscillatory activity were evaluated based on the phase synchrony between the motor (C3) and visual (Oz) electrodes, calculated using the phase-locking value (PLV) [25]. The PLV for an electrode pair (*e_i_*, *e_j_*) was calculated as follows:$$PLV\left(t,f,e_{i},e_{j}\right)=\frac{1}{N}\left|\sum_{n=1}^{N}{exp}^{i\left(\phi \left(t,f,n,e_{i}\right)-\phi \left(t,f,n,e_{j}\right)\right)}\right|$$
where N represents the number of trials. The normalized PLVs were computed as follows:PLVt,f,ei,ejnorm=PLVt,f,ei,ej−μbaseσbase
where *µ_base_* and *σ_base_* are the mean and standard deviation of PLVs over a 300 ms pre-stimulus baseline, respectively.

Due to the scarcity of data after the removal of outliers, the PLVs were calculated as a single value per condition using all EEG trials across participants; thus, individual z-values could not be derived. Therefore, total z-values were calculated using the individual PLVs to determine whether an increase in the poststimulation session of PLVs occurred across participants using the Wilcoxon signed-rank test. Positive z-values indicated that the PLVs were greater in the post-stimulation session than in the pre-stimulation session.

The correlations between the differences in RTs and PLVs between the pre- and post-stimulation sessions for all conditions were analyzed using Spearman’s correlation coefficient to determine the relationship between the stimulation effects on motor performance and the modulatory effects of stimulations on the sensorimotor network. Statistical differences were evaluated using the Mann–Whitney U test.

## 3. Results

### 3.1. Motor Performance

The in-stimulation session RTs were significantly shorter than the baseline session RTs (*p* < 0.05) in eight, three, four, three, and two participants during motor-anodal tDCS, visual-anodal tDCS, alpha-tACS, beta-tACS, and sham conditions, respectively. Similarly, the post-stimulation session RTs were significantly shorter than the baseline session RTs (*p* < 0.05) in seven, three, four, two, and two participants during motor-anodal tDCS, visual-anodal tDCS, alpha-tACS, beta-tACS, and sham conditions, respectively. The average z-values for each comparison and condition are shown in Figure 2.

Among conditions, the in-stimulation session RTs were significantly shorter than the baseline session only in the motor-anodal tDCS (z = −3.55; *p* < 0.01; r = −0.36) condition but not in the visual-anodal tDCS (z = −0.48; *p* = 0.63; r = −0.16), alpha-tACS (z = −0.80; *p* = 0.43; r = −0.20), beta-tACS (z = −0.39; *p* = 0.70; r = −0.15), and sham (z = −0.19; *p* = 0.85; r = −0.03) conditions. Similarly, the post-stimulation session RTs were significantly shorter than the baseline session in the motor-anodal tDCS condition (z = −2.77; *p* < 0.01; r = −0.39) but not in the other conditions (visual-anodal tDCS [z = −0.41; *p* = 0.68; r = −0.09], alpha-tACS [z = −0.95; *p* = 0.34; r = −0.19], beta-tACS [z = −0.17; *p* = 0.86; r = −0.14], and sham [z = −0.34; *p* = 0.73; r = −0.05]).

The average RTs (in seconds) for each condition across sessions were as follows: In the pre-stimulation session, RTs were 0.432, 0.412, 0.409, 0.399, and 0.395 s for motor-anodal tDCS, visual-anodal tDCS, alpha-tACS, beta-tACS, and sham, respectively. During the in-stimulation session, RTs were 0.417, 0.411, 0.388, 0.393, and 0.394 s, respectively, and, in the post-stimulation session, 0.407, 0.415, 0.402, 0.392, and 0.395 s, respectively.

Accuracy remained high across all conditions, with minimal variation between sessions. In the pre-, in-, and post-stimulation sessions, accuracy was 98.57%, 98.96%, and 98.37% for motor-anodal tDCS; 97.85%, 97.14%, and 97.40% for visual-anodal tDCS; 97.14%, 97.79%, and 97.14% for alpha-tACS; 97.27%, 96.09%, and 97.40% for beta-tACS; and 98.44%, 98.31%, and 98.31% for sham, respectively.

### 3.2. Oscillatory Activity

#### 3.2.1. Alpha and Beta Amplitudes

Under all conditions, no significant difference was observed in the alpha (10 Hz) amplitude between the pre- and post-stimulation sessions. In the motor-anodal tDCS condition, significant increases in the beta (20 Hz) amplitude were observed in both the motor (C3; z = 3.55; *p* < 0.01) and visual areas (Oz; z = 2.72; *p* < 0.01). Specifically, significantly positive z-values (*p* < 0.05) were observed in six and five participants for the C3 and Oz electrodes, respectively. No significant differences were found between the pre- and post-stimulation session in other conditions (visual-anodal tDCS [motor amplitude, z = 0.02, *p* = 0.98; visual amplitude, z = 0.98, *p* = 0.33], alpha-tACS [motor amplitude, z = 0.86, *p* = 0.39; visual amplitude, z = 0.65, *p* = 0.51], beta-tACS [motor amplitude, z = 0.90, *p* = 0.38; visual amplitude, z = 1.45, *p* = 0.15], and sham [motor amplitude, z = 0.68, *p* = 0.49; visual amplitude, z = 0.67, *p* = 0.50]).

The average z-values for each condition and the corresponding amplitudes are shown in Figure 3.

For additional insight into the overall spectral profile, a full-band spectrogram is provided as Appendix A.

#### 3.2.2. Phase Synchrony Between the Motor and Visual Areas

Among the conditions, a significant increase in beta (i.e., 20 Hz)-PLVs was observed only under the motor-anodal tDCS condition (z = 2.96; *p* < 0.01) but not under the other conditions (visual-anodal tDCS [z = 1.41; *p* = 0.16], alpha-tACS [z = 0.99; *p* = 0.32], beta-tACS [z = 0.76; *p* = 0.45], or sham [z = 1.77; *p* = 0.45] conditions). The differences in the beta-band PLVs between the pre- and post-stimulation sessions for each condition are shown in Figure 4. In contrast, no significant difference was observed in the alpha-band (i.e., 10 Hz) PLVs between the pre- and post-stimulation sessions for any condition.

#### 3.2.3. Correlation Between Pre- and Post-Differences in Motor Performance and EEG Oscillatory Activity

No statistically significant correlations were found between pre- and post-differences in EEG amplitudes and RTs across all conditions. However, a significant negative correlation was observed between the pre- and post-differences in RTs and beta-PLVs under the motor-anodal tDCS condition (r = −0.78, *p* < 0.03). No such correlations were observed in the other conditions: visual-anodal tDCS (r = −0.31, *p* = 0.45), alpha-tACS (r = 0.57, *p* = 0.14), beta-tACS (r = −0.26, *p* = 0.54), and sham (r = −0.60, *p* = 0.12), as shown in Figure 5. Additionally, no significant correlations were found between the pre- and post-differences in RTs and alpha-PLVs.

## 4. Discussion

This study aimed to enhance motor performance using tES and to investigate its modulatory effects on local and global beta oscillatory activity relevant to tasks and stimulation. The results of this preliminary study suggest that motor-anodal tDCS may enhance motor performance, as evidenced by significantly shorter RTs during and after stimulation than at baseline. While motor-anodal tDCS led to a significant increase in beta amplitude in the motor area during the pre- and post-stimulation sessions, the observed performance improvements were not directly correlated with this change. Instead, these enhancements appeared to be associated with increased beta-phase synchronization between the motor and visual areas.

Interestingly, tDCS modulated the behavior only when the anodal and cathodal electrodes were placed on the motor and visual areas, respectively. This design was chosen because our objective was not only to assess local amplitude changes at Oz and C3 but also to evaluate the connectivity between these regions via phase synchronization. By configuring the montage with the anodal electrode over one region and the cathodal over the other, we modeled a current flow between Oz and C3. Notably, increased phase synchronization and faster reaction times were observed only when the anodal electrode was positioned over C3, suggesting that excitatory stimulation of the motor cortex plays a key role in enhancing motor performance. This electrode configuration suggests that current flow from the visual (cathodal) to the motor (anodal) areas may facilitate neuronal signaling, enhancing the directional flow of information from visual inputs to motor outputs. Such facilitation could underpin the observed improvements in visual-motor responses. These findings highlight the importance of considering the electrode placement and directionality of current flow when designing tDCS interventions to modulate task-relevant networks. However, the relationship between local synchronization, reflected in the oscillatory amplitude or power in individual areas, and global synchronization, represented by beta-phase connectivity between areas, remains unclear. Although this study suggests that global synchronization plays a prominent role in enhancing motor performance, the interaction between local and global oscillatory dynamics warrants further investigation.

These findings imply that the effects of motor-anodal tDCS on performance might not stem primarily from the localized modulation of neuronal activity but rather from its influence on global network connectivity, potentially facilitating more efficient information processing within the sensorimotor network. Beta-phase synchronization likely plays a key role in integrating visual and motor information, thereby improving task-related responses. tDCS has been shown to transiently increase the small-world properties of brain connectivity, reflecting the enhanced efficiency of neural communication across the cortex [26]. This aligns with the notion that tDCS enhances motor performance by modulating global network dynamics, rather than solely affecting localized brain regions. Moreover, this interpretation aligns with our previous TMS-EEG study [27], which indicated that electroconvulsive therapy, an invasive neuromodulation technique, may alleviate psychiatric symptoms by enhancing information processing across neural networks, as assessed through PLVs.

The observed reduction in RTs and positive correlation with increased visual-motor beta-phase synchronization underscores the potential role of beta synchronization in facilitating information transfer and processing within the sensorimotor network. Furthermore, accuracy was maintained at consistently high levels across all conditions, suggesting that the reduction in RTs was not accompanied by any loss in response precision. These results are consistent with those of prior studies, indicating that sensory-motor beta oscillations are critical for motor preparation, execution, and task-specific information processing [3,4,28].

Contrary to our expectations, this study did not observe a significant enhancement of tACS on motor performance. As a result, we could not establish a clear relationship between the motor enhancement effects of tACS and beta oscillatory activity. One possible explanation for this is that the tACS protocol implemented in this study may not have been sufficient to effectively modulate brain activity and motor performance. This outcome is consistent with the mixed findings in the literature, as some studies have reported difficulties in achieving measurable behavioral improvements such as enhanced motor performance through tACS [29].

These findings highlight the inherent complexity of inducing motor performance enhancement via tACS and underscore the need for more refined stimulation approaches in future research. To better understand the mechanisms underlying tACS and its potential effects, future studies should consider strategies to enhance the precision and efficacy of stimulation. These approaches include more accurate electrode placement using co-registration with TMS or fMRI, tailoring stimulation to individual frequency characteristics, optimizing phase angles, and extending the duration of stimulation. By addressing these factors, future studies could elucidate the conditions under which tACS can effectively modulate brain activity and improve motor performance.

The present study has several limitations. First, the sample size was small (*n* = 8), which limits the generalizability of the findings and reduces the statistical power of group-level analyses. A larger sample size would provide more robust insights into the effects of tDCS and tACS on motor performance and oscillatory activity, accounting for individual variability and enhancing the reliability of the findings.

Second, individual differences in response to tACS and tDCS may be attributable to variability in the participants’ peak beta frequencies, which ranged from 18 to 24 Hz. This finding suggests that a uniform stimulation protocol may not be optimal for all individuals. Future studies should consider tailoring tACS and tDCS parameters, such as stimulation frequencies and phase angles, to individual characteristics to maximize efficacy.

Thirdly, this study did not formally assess blinding effectiveness. Participants were informed they would receive brain stimulation but were unaware of different stimulation types or the sham condition, minimizing expectancy biases. To enhance blinding, the sham condition included a 30 s ramp-up and ramp-down, a standard method for mimicking active stimulation. However, we did not collect post-experiment feedback on condition perception, and experimenters were aware of stimulation assignments, which may introduce bias. Future studies should consider post-experiment blinding assessments and double-blind designs to improve methodological rigor.

Finally, the limited number of task trials in this study may have constrained the sensitivity of the analyses. Increasing the number of trials in future research could help capture the more nuanced effects of stimulation on motor performance and brain activity.

Future studies should address these limitations to refine our understanding of the mechanisms underlying the effects of tACS and tDCS on motor performance and neural oscillations. These efforts will contribute to the development of more effective neuromodulatory interventions for enhancing motor function.

## Figures and Tables

**Figure 1 brainsci-15-00286-f001:**
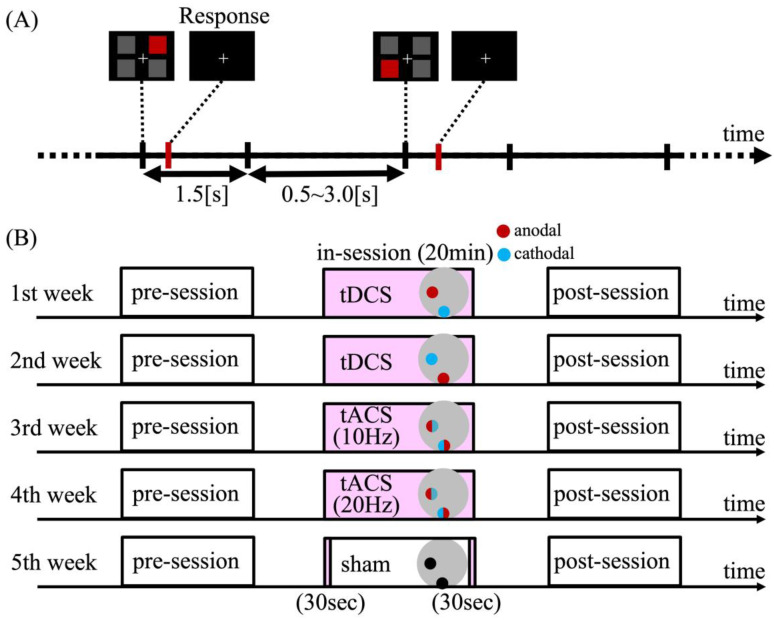
(**A**) Standard trial outline. Participants are instructed to detect a red square among four squares on a computer display and to press an appropriate key on a keyboard following detection of the stimulus. (**B**) Experimental procedure. Each participant completes a set of experiments consisting of five stimulation conditions. Each experiment consists of 3 sessions (pre-stimulation session, in-stimulation session, and post-stimulation session), and each session consists of two blocks of trials. Inter-condition intervals of at least 1 week from the previous condition.

**Figure 2 brainsci-15-00286-f002:**
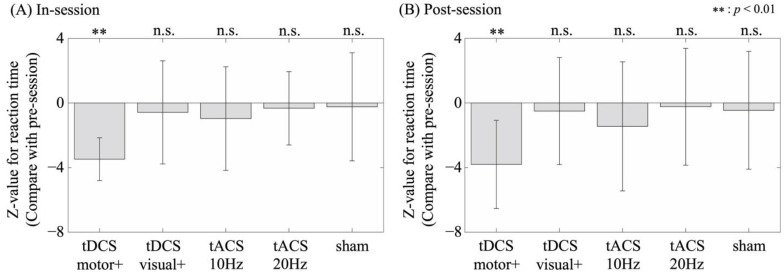
Comparison of participant-averaged z-values for RTs: (**A**) between the pre/in-stimulation session and (**B**) between the pre/post-stimulation session in the motor-anodal tDCS (tDCS motor+), visual-anodal tDCS (tDCS visual+), alpha-tACS (tACS 10Hz), beta-tACS (tACS 20Hz), and sham condition. Error bars denote standard deviations.

**Figure 3 brainsci-15-00286-f003:**
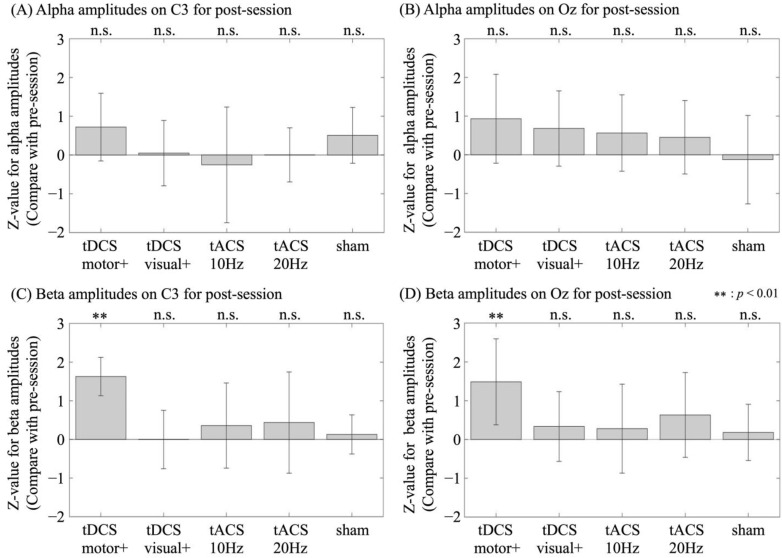
Comparison of participant-averaged z-values for amplitude changes in the pre/post-stimulation session across conditions: for alpha amplitude change on (**A**) C3 (motor area) and (**B**) Oz (visual area) and for beta amplitude changes on (**C**) on C3 (motor area) and (**D**) Oz (visual area), in the motor-anodal tDCS (tDCS motor+), visual-anodal tDCS (tDCS visual+), alpha-tACS (tACS 10 Hz), beta-tACS (tACS 20 Hz), and sham condition. Error bars denote standard deviations.

**Figure 4 brainsci-15-00286-f004:**
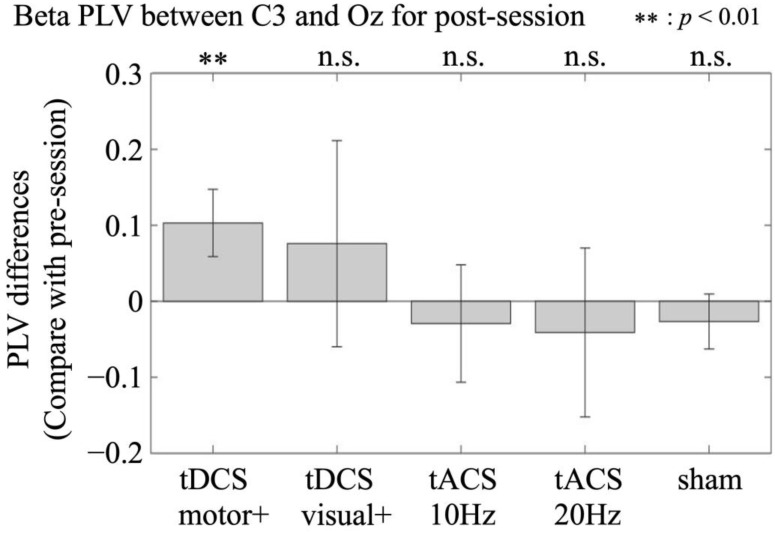
Participant-averaged beta C3-Oz PLV differences between the post- and the pre-stimulation session in the motor-anodal tDCS (tDCS motor+), visual-anodal tDCS (tDCS visual+), alpha-tACS (tACS 10 Hz), beta-tACS (tACS 20 Hz), and sham condition. Error bars denote standard deviations.

**Figure 5 brainsci-15-00286-f005:**
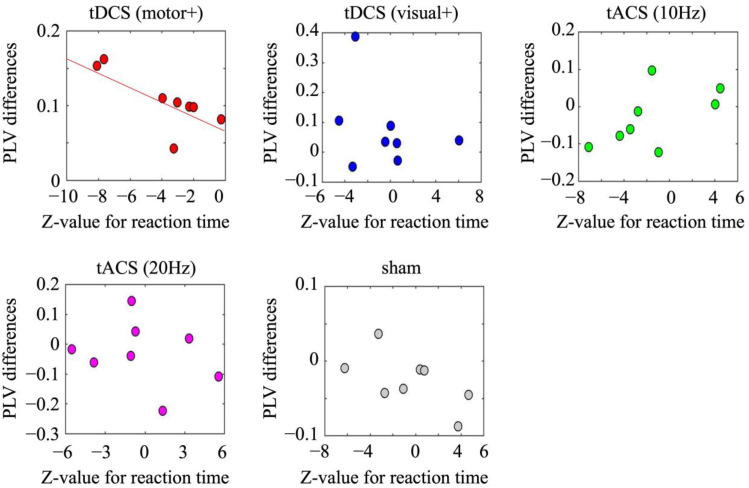
Scatter plots between z-values for RTs and beta-PLV differences between the pre/post-stimulation session in the motor-anodal tDCS (tDCS motor+), visual-anodal tDCS (tDCS visual+), alpha-tACS (tACS 10 Hz), beta-tACS (tACS 20 Hz), and sham condition. Error bars denote standard deviations.

## Data Availability

The preliminary data presented in this study are available upon request from the corresponding author.

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
