# Peer review of "Transcranial Direct Current Stimulation Enhances Motor Performance by Modulating Beta-Phase Synchronization in the Sensorimotor Network: A Preliminary Study"

_brainsci, 2025, doi:10.3390/brainsci15030286_

Round 1

Reviewer 1 Report

Comments and Suggestions for Authors

Miyauchi et al. used EEG to evaluate the effects of tDCS and tACS on beta-PLV within the sensorimotor network and observed current-specific effects of the transcranial stimulation as well as neurobehavioral correlation. I have the following major concerns.

Major:

The sample size is too small to infer interpretations. Was the power analysis done? Or else, report effect sizes.

Please specify why were tDCS and tACS compared, considering they have very different mechanisms of action.

What was the rationale for using alpha tACS?

Please indicate ramp-up or ramp-down periods of stimulation.

How effective was the blinding?

Please provide a current flow/density modeling to show the spatial distribution of stimulation.

The study design does not reliably answer the author’s research question. They specify that for motor stimulation, an anodal electrode was placed on C3, whereas for visual, it was placed on the visual area, which indicates that a cathodal electrode was used as a reference on the Oz for motor anodal-tDCS and on the motor region for visual anodal-tDCS. Cathodal tDCS has been shown by a large volume of studies to cause neuromodulation as well, which means it’s an active stimulation montage and is placed on the cortex, it cannot be considered a reference. How can authors be sure that the observed effects are following anodal stimulation and are not confounded by cathodal stimulation?

Stimulation has been shown to cause changes in the baseline power. Why were pre-stimulation data considered as a baseline? It doesn’t account for the stimulation or other factors affecting it (e.g., the effect of timing) thus making the baseline highly unreliable for comparison. Why wasn’t the inter-trial interval used as a baseline instead?

Soon after the target onset, participants responded by pressing a button and proceeded on to the next trial. As it has been shown multiple times in the literature, a movement (i.e., a button press) is associated with peri-movement beta desynchronization followed by beta rebound activity (PMBR), which lasts about 1500-2000ms after movement onset. The inter-trial interval was 500-3000ms, However, with it less than 2000 ms, there would be overlapping effects, which questions the integrity of the neural activity of interest.

What were the average RTs?

Why isn’t the accuracy analyzed/reported? Could the decrease in RT be at the expense of accuracy?

Please provide spectrograms showing neural activities.

Author Response

Dear Reviewer, 

Thank you for your through review and comments. 

Please find attached the PDF file for our reply.

Reviewer 2 Report

Comments and Suggestions for Authors

General comments

This manuscript aims to explore whether transcranial direct current stimulation (tDCS) and alternating current stimulation (tACS) can enhance sensorimotor responses by modulating beta-band synchronisation. The authors’ aim is commendable. The authors found significant reductions in response time (RT) only after motor-anodal tDCS. Electroencephalography analysis shows a positive correlation between these RT reductions and increased beta-phase synchronisation between visual and motor areas. In contrast, tACS does not lead to significant RT improvements or changes in beta-phase synchronisation. The authors fulfil their aim sufficiently despite some specific and minor issues detailed below.

Specific comment

There is no true control condition, viz., neither stimulation nor sham.

Minor comments

(line 13) … stimulation (tACS) could…

(l19 and elsewhere throughout manuscript, as well) please, do not start sentences with acronyms;

(l142 and elsewhere throughout manuscript, as well) please, do not use acronyms in headings;

(l182) please, improve equations’ legibility.

Author Response

(The authors gave the same response as above.)

Round 2

Reviewer 1 Report

Comments and Suggestions for Authors

The authors have reasonably responded to my comments.